**Data Availability Statement:** Data Availability Statement: The data that support the findings of this study are available from the Italian regions participating to MoM-Net group (Lombardy,

# Monitoring medicine prescriptions before, during and after pregnancy in Italy

**Filomena Fortinguerra**[ID][1]*, **Valeria Belleudi**[2], **Francesca Romana Poggi**[2], **Serena Perna**[1], **Renata Bortolus**[3], **Serena Donati**[4], **Paola D'Aloja**[4], **Roberto Da Cas**[5], **Antonio Clavenna**[6], **Anna Locatelli**[ID][7], **Antonio Addis**[2], **Marina Davoli**[2], **Francesco Trotta**[1], **MoM-Net group**[¶]

**1** Italian Medicines Agency (AIFA), Rome, Italy, **2** Department of Epidemiology, Lazio Regional Health Service, Rome, Italy, **3** Directorate General for Preventive Health–Office 9, Ministry of Health, Rome, Italy, **4** National Centre for Disease Prevention and Health Promotion, Istituto Superiore di Sanità –Italian National Institute of Health, Rome, Italy, **5** Pharmacoepidemiology Unit, National Centre for Drug Research and Evaluation, Istituto Superiore di Sanità –Italian National Institute of Health, Rome, Italy, **6** Laboratory for Pharmacoepidemiology, Department of Public Health, IRCCS–Istituto di Ricerche Farmacologiche Mario Negri, Milan, Italy, **7** Department of Obstetrics and Gynecology, University of Milano-Bicocca, Monza, Italy

¶ Membership of MoM-Net group is provided in the Acknowledgments.
* f.fortinguerra@aifa.gov.it

## Abstract

### Background

The use of medications during pregnancy is a common event worldwide. Monitoring medicine prescriptions in clinical practice is a necessary step in assessing the impact of therapeutic choices in pregnant women as well as the adherence to clinical guidelines. The aim of this study was to provide prevalence data on medication use before, during and after pregnancy in the Italian population.

### Methods

A retrospective prevalence study using administrative healthcare databases was conducted. A cohort of 449,012 pregnant women (15–49 years) residing in eight Italian regions (59% of national population), who delivered in 2016–2018, were enrolled. The prevalence of medication use was estimated as the proportion (%) of pregnant women with any prescription.

### Results

About 73.1% of enrolled women received at least one drug prescription during pregnancy, 57.1% in pre-pregnancy and 59.3% in postpartum period. The prevalence of drug prescriptions increased with maternal age, especially during the 1st trimester of pregnancy. The most prescribed medicine was folic acid (34.6%), followed by progesterone (19%), both concentrated in 1st trimester of pregnancy (29.2% and 14.8%, respectively). Eight of the top 30 most prescribed medications were antibiotics, whose prevalence was higher during 2nd trimester of pregnancy in women ≥ 40 years (21.6%). An increase in prescriptions of antihypertensives, antidiabetics, thyroid hormone and heparin preparations was observed during pregnancy; on the contrary, a decrease was found for chronic therapies, such as antiepileptics or lipid-modifying agents.

Veneto, Emilia Romagna, Tuscany, Umbria, Lazio, Puglia, Sardinia) but restrictions apply to the availability of these data, which were used under license (as by third-party sources) for the current study, and so are not publicly available. However, data are available from the authors with permission of Italian regions, which are the data owner. The non-author contact information to which data requests may be sent is: ufficio.osmed@aifa.gov.it.

**Funding:** The author(s) received no specific funding for this work.

**Competing interests:** The authors have declared that no competing interests exist.

## Conclusions

This study represents the largest and most representative population-based study illustrating the medication prescription patterns before, during and after pregnancy in Italy. The observed prescriptive trends were comparable to those reported in other European countries. Given the limited information on medication use in Italian pregnant women, the performed analyses provide an updated overview of drug prescribing in this population, which can help to identify critical aspects in clinical practice and to improve the medical care of pregnant and childbearing women in Italy.

## Introduction

The use of medications during pregnancy is a common event, which has increased in recent years worldwide. In high income countries the prevalence of women who received at least one medication during pregnancy ranges from 27% to 99% [1, 2].

The medication use during pregnancy is a topic of great interest in the field of public health, due to the possible adverse outcomes that this event can have on the health status, even in the long term, of both women and newborns. Although most drugs are able to cross the placenta and reach the embryo and fetus [3], only few of them can cause fetal malformations or alter the normal development of the fetus, generally occurring as a consequence of chronic therapies or long term treatments [4, 5]. It is also well-known that pharmacological treatments commonly used in pregnancy are often not sufficiently tested in this population; limited data on their pharmacokinetics and safety profile (especially on long-term adverse effects) are available, and thus, they are prescribed without an adequate evidence to evaluate their risks and benefits in pregnant women. Even patient information leaflets are not useful in guiding clinician's prescribing choices [6–9]. This denies pregnant women appropriate and adequate drug therapies potentially compromising both the mother's and the child's health. Furthermore, since the pregnancy itself may influence maternal chronic diseases, where medication use is essential to both maternal and fetus health, a better knowledge regarding the potential impact of the pregnancy on the efficacy of the maternal chronic pharmacological therapies is needed [10, 11]. In this perspective, in 2020 the Italian Medicines Agency (AIFA) promoted the creation of an Italian network, called MoM-Net (Monitoring Medication Use During Pregnancy Network), which work as a national observatory for monitoring medication use in the Italian pregnant population over time [12]. Actually, it includes eight Italian regions and experts from public or academic Italian institutions and is eventually extendable to other regions.

Observational and/or descriptive studies on medication use in pregnant women [13–15] coordinated at national level, can help to fill important information gap in maternal and perinatal medicine to improve the Italian clinical practice in treatment choices in pregnancy, but also to promote interventions to reduce intra-regional and inter-regional variability in prescription patterns. Furthermore, MoM-Net may work as a surveillance system able to identify inappropriate or unsafe prescriptions in pregnancy and to provide timely answers to any emerging questions regarding medication use during the perinatal period.

The first study conducted by the network consisted in a large population-based analysis investigating drug prescriptions before, during and after pregnancy [16]. The project represents an important novelty in the national panorama, since to date the available data on drug prescription during pregnancy were not updated [17–19] or still limited to the experience of few regions [20–26]. The aim of this article was to describe the results of overall analysis on

drug prescriptions in a representative sample of the Italian pregnant women population before, during and after pregnancy.

## Methods

### Selection of the study population

A retrospective population study using administrative healthcare databases was performed on a cohort of women aged between 15 and 49 years, who gave birth (live births and stillbirths) in a Maternity Unit from April 1st, 2016 to March 31th, 2018 and who were resident in eight Italian regions: Lombardy, Veneto, Emilia-Romagna in the North, Tuscany, Umbria, Lazio in the Centre, Puglia and Sardinia in the South. The selected population represented the 5.9% of the Italian population of childbearing age (15–49 years) in the period 2016–2018 (an average of 7.6 millions of people) and the 59% of all deliveries occurred in Italy in the study period. Only women covered by the Italian National Healthcare Service in the study period were selected. In the case of women with more than one birth during the study period, only the first birth was included in the study.

The need for participants consent was waived by the ethics committee because it is not needed in Italy in case of descriptive study, based on retrospective data routinely collected by local structures of Italian regional health authorities, which are kept properly anonymised within administrative healthcare databases, with no need to obtain an informed consent at the time of original data collection.

### Sources of data

Pregnant women were identified through the Regional Birth Registry (CeDAP, *Certificato di Assistenza al Parto*), while the data on prescriptions of medicines reimbursed by the National Healthcare Service (NHS) were retrieved from the Regional Drug Prescription Database. Voluntary abortions and miscarriages (pregnancy loss before 180 days of amenorrhea) were not included in the study, as this information were not recorded in the CeDAP database [27].

The data from the two different health care databases were combined using a deterministic record-linkage procedure based on anonymized personal identification codes, which is a procedure in line with privacy legislation [28].

### Definition of the study period

The start date of pregnancy was estimated as the difference between the date of birth and the gestational age at birth expressed as days (calculated by multiplying the number of weeks of amenorrhea reported in the CEDAP database by 7 days). The following three-time windows were identified:

- the **pre-pregnancy period**, defined as 3 trimesters before the last menstrual period (LMP) date (273 days before LMP date);

- the **pregnancy period**, including the *I trimester of pregnancy* (1st TP) defined as the period between 0 (LMP date = start date of pregnancy) and the day 91 following the start date of pregnancy; the *II trimester of pregnancy* (2nd TP), defined as the period between the day 92 and the day 189 from the start date of pregnancy (or date of birth if the birth occurred during the 2nd TP, which is within 27 weeks of gestation); the *III trimester of pregnancy* (3rd TP) defined as the period between the day 190 from the start of pregnancy and the date of birth;

- the **post-pregnancy period** defined as 3 trimesters after the date of birth (273 days following the date of birth);

For each woman enrolled in the study, the socio-demographic characteristics (e.g. age, nationality, education, and occupational status), the clinical information related to pregnancy (e.g. gestational age and parity) and obstetric history of the pregnant women (e.g. previous deliveries, previous cesarean sections, and previous abortions) were retrieved from the CeDAP database.

### Definition of drug prescription and users

The pharmaceutical prescriptions retrieved from the regional Drug Prescription Databases provided in the identified periods were linked to selected pregnant women cohort. The medicines investigated were classified according to the World Health Organization (WHO) Anatomical Therapeutic Chemical (ATC coding) classification system [29]. Medication dispensing records with missing information (name of medication, dispensing date) were excluded from the analysis.

The prevalence of medication use was estimated as the proportion (%) of pregnant women who received at least one prescription of any medication in the time window of interest (pre, in, post pregnancy period or trimester).

The results are expressed as prevalence of use by ATC I level, therapeutic categories and drug. Three therapeutic categories were included in the analysis as follows: drugs linked to the condition of pregnancy (e.g. vitamins, minerals and anti-anemic preparations), drugs for acute disorders (e.g. antibiotics) and drugs for pre-existing chronic diseases or diagnosed for the first time in pregnancy (e.g. antihypertensives). The prevalence of the main therapeutic categories and the ranking of the first thirty drugs used in pregnancy were analyzed by maternal age group and period or trimester before, during and after pregnancy.

### Results

Data on medication prescriptions in a cohort of 449,012 women aged between 15 and 49 years who became pregnant between 1 April 2016 and 31 March 2018 were included in the analysis (**Table 1**). About 37.5% of selected pregnant women were aged $\geq$ 35 years, 9.7% were at least 40 years old, the 0.8% was 45 years old and over. The 19.8% of the pregnant women were foreign citizens, coming mainly from high-developed countries and 23.8% had a primary school degree. Almost 50.7% were primiparous, while the 19.7% experienced at least one previous abortion (5% of these had $\geq$2 abortions). The majority of the pregnancies (92.5%) ended at term, while the 6.9% ended before 37 weeks. The caesarean section rate was 30.3%, assisted reproductive technology (ART) regarded the 3% of pregnancies and 11.9% of women underwent invasive prenatal diagnosis. The 1.8% of the pregnancies were multiple.

Overall, 328,347 of pregnant women (73.1% of the selected study population) received at least one drug prescription during pregnancy, 57.1% in the pre-pregnancy period and 59.3% in the postpartum period. When all the trimesters are analysed separately (**Fig 1**), the prevalence trend of pregnant women who received at least one drug prescription was stable during the three pre-pregnancy trimesters, ranging from 31.5% to 33.0%, then it increased during 1st TP reaching the peak of 51.5%; decreased up to 41.6% in 3rd TP and raised again to 45.6% in the first trimester post-partum. In the second and third trimesters after birth the prevalence decreased up to a lower level than that observed in the pre-pregnancy period (around 20%).

During all pregnancy trimesters, as well as during the trimesters before and after pregnancy, the prevalence of drug prescriptions increased with maternal age, especially during the 1st TP (**Fig 2**) describes drug prescriptions trends by ATC group before, during and after pregnancy. The drugs for blood and haemopoietic organs (ATC B) were the most prescribed medications during pregnancy, with a peak of prevalence in 1st TP (32.5%). They were followed by systemic

**Table 1. Study cohort characteristics (n = 449,012).**

| | n. | % |
|---|---:|---|
| **Age group** | | |
| ≤ 24 | 33,651 | 7.5 |
| 25–29 | 92,333 | 20.6 |
| 30–34 | 154,588 | 34.4 |
| 35–39 | 124,680 | 27.8 |
| 40–44 | 40,322 | 8.9 |
| ≥ 45 | 3,438 | 0.8 |
| **Nationality** | | |
| Italian | 358,467 | 79.8 |
| Foreign | 88,629 | 19.8 |
| *High-income countries* | *86,159* | *97.7* |
| *Low-income countries* | *2,470* | *2.3* |
| *Missing* | *1,916* | *0.4* |
| **Level of education** | | |
| None/primary school | 106,759 | 23.8 |
| Secondary school | 200,618 | 44.7 |
| Bachelor degree/post-bachelor degree | 139,559 | 31.1 |
| *Missing* | *2,076* | *0.4* |
| **Occupational status** | | |
| Employed | 284,069 | 63.3 |
| Unemployed/Looking for first job | 54,492 | 12.1 |
| Housewife | 98,450 | 21.9 |
| Other | 7,210 | 1.6 |
| *Missing* | *4,791* | *1.1* |
| **Previous birth** | | |
| No | 227,525 | 50.7 |
| Yes | 221,487 | 49.3 |
| *of which cesarean section* | *59,782* | *27.0* |
| **Previous miscarriage** | | |
| 0 | 360,619 | 80.3 |
| 1 | 65,997 | 14.7 |
| 2 | 22,396 | 5.0 |
| **Gestational age at birth** | | |
| <37 weeks | 30,774 | 6.9 |
| 37–41 weeks | 415,366 | 92.5 |
| >41 weeks | 2,872 | 0.6 |
| **Parity** | | |
| 1 | 440,765 | 98.2 |
| 2+ | 8,247 | 1.8 |
| **Invasive prenatal diagnosis** | | |
| None | 394,785 | 87.9 |
| Chorionic villus sampling | 20,435 | 4.6 |
| Amniocentesis | 31,423 | 7.0 |
| Other invasive test | 1,433 | 0.3 |
| *Missing* | *936* | *0.2* |
| **Assisted reproductive technology (ART)** | | |
| No/*not classified* | 360,558 | 97.0 |

*(Continued)*

**Table 1.** (Continued)

|  | n. | % |
|---|---|---|
| Yes | 11,233 | 3.0 |
| **Cesarean section in the index pregnancy** |  |  |
| No | 312,785 | 69.7 |
| Yes | 136,227 | 30.3 |

antimicrobial (ATC J) with a peak in 2nd TP (16.4%), and by drugs for the genito-urinary system and sex hormones (ATC G), with a peak of 15.9% in 1st TP, decreasing progressively up to 3.9% in 3rd TP. With respect to systemic antimicrobials, a high steady-state level of prescription was also observed in pre-pregnancy as well as in the post-partum period, where the prevalence ranged from 10% to 15%.

For all categories considered, the prevalence of drug use by maternal age group (Table 2) showed a growing trend as the maternal age increased regardless of the period (pre, in and post-pregnancy).

The only exception was represented by anti-anaemic preparations (ATC B03), which showed the highest prevalence in 1st TP and remaining almost stable across all age groups.

Consistently with the ATC I level, the use of antibiotics (ATC J01) is particularly high for all age groups, not only during pregnancy, but also in the pre-pregnancy and post-pregnancy trimesters, with the higher prevalence during 2nd TP in women aged 40 and over when compared with younger women. Considering the category of progestogens (ATC G03) the highest prevalence was observed in 1st TP in women aged 40 and over.

A high level heparin preparations (ATC B01AB) was observed among women 40 years old and over during the first trimester of post-partum period and during the 3rd TP.

According to the therapeutic indication, gonadotropins showed the highest prescription level in the trimester immediately before the start of pregnancy, with an increasing trend with maternal age.

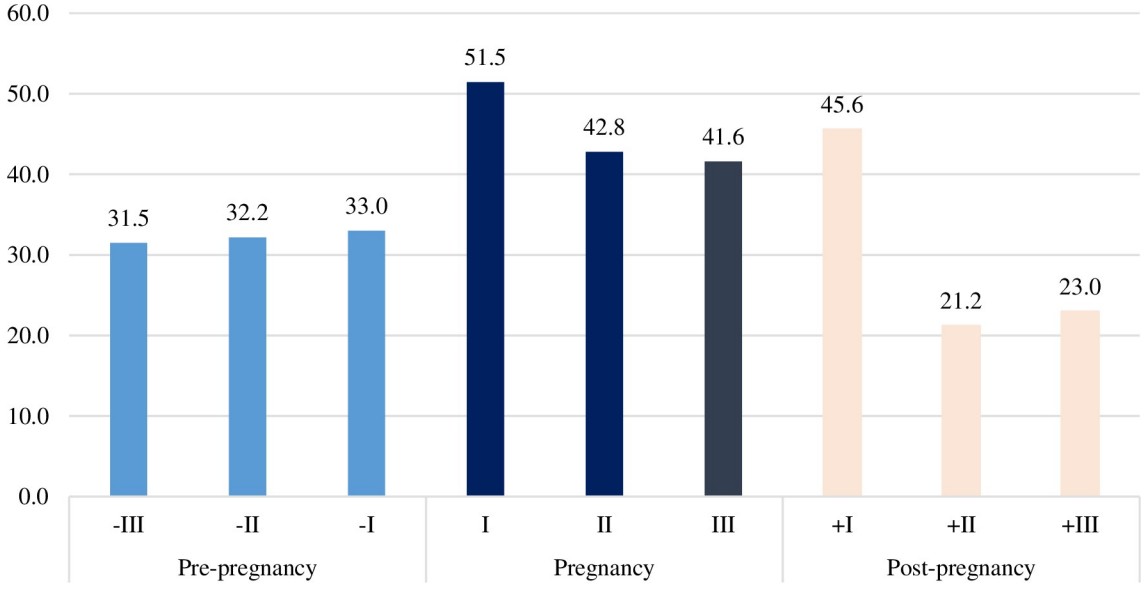

**Fig 1. Drug prescriptions by trimester before, during and after pregnancy.** *denominator: pregnancies reaching the 3rd TP (exclusion of deliveries occurred between 20–27 weeks of gestation).

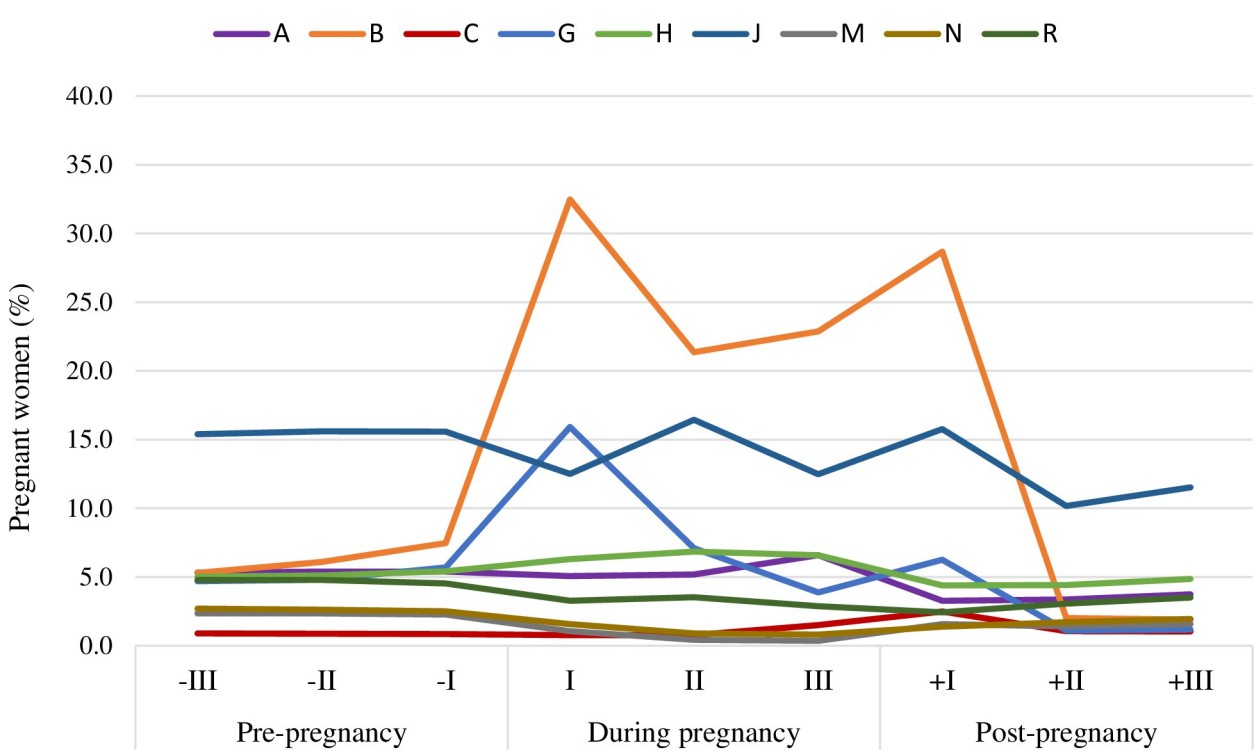

**Fig 2. Drug prescriptions trends by ATC group (I level)\* before, during and after pregnancy.** \* ATC groups with prevalence during pregnancy <0,5% were excluded.

For other therapeutic categories the prescribing patterns across trimesters were similar in different age groups, with a decreasing prevalence in pregnancy for antiepileptics, lipid-modifying agents, anti-inflammatory drugs, and increasing prevalence for corticosteroids in 1st TP in women aged ≥40, thyroid hormone (peak in 2nd TP), antidiabetics (peak in 3rd TP) and antihypertensive (peak in the first postpartum trimester).

The most prescribed medications during pregnancy (**Table 3**) were folic acid (34.6%), progesterone (19.0%), ferrous sulphate (18.8%) and amoxicillin/clavulanic acid (11.5%). The first two medications were mainly concentrated in the 1st TP and decreased significantly in the 2nd and 3rd TP. It should be noted that eight of the top 30 most prescribed medications during pregnancy were antibiotics for systemic use (fosfomycin, azithromicyn, amoxicillin, amoxicillin/clavulanic acid, cefixime, clarithromycin, ampicillin, ciprofloxacin). **Fig 3** highlights wide age-related prescribing variations, fosfomycin was mostly prescribed in younger women <34 years old, while azithromycin, was most commonly prescribed in women 40 years old and over.

## Discussion

This study showed that pregnancy medication use is very common in Italy. Seven in 10 pregnant women received at least one drug prescription during pregnancy. The prevalence of overall medication prescriptions throughout the study period varied by trimester of pregnancy and by woman's age. We found an increased medication use during the 1st TP (51.5%), and with increasing maternal age (64.7% ≥ 40 years). A previous analysis, based on prescription data in a smaller Italian sample showed about 80.4% of women exposed to at least one prescription medication during pregnancy, with a similar trend across all trimesters [20].

**Table 2. Prevalence (%) by therapeutic group and maternal age before, during and after pregnancy.**

| Therapeutic groups | Age group | Pre-pregnancy | | | During pregnancy | | | Post-pregnancy | | |
|---|---|---|---|---|---|---|---|---|---|---|
| | | -III | -II | -I | I | II | III | +I | +II | +III |
| Vitamins and minerals supplements | **all** | **1.0** | **1.0** | **1.2** | **1.3** | **1.5** | **1.3** | **0.9** | **1.0** | **1.1** |
| | ≤ 34 | 0.7 | 0.8 | 0.9 | 1.0 | 1.2 | 1.0 | 0.7 | 0.8 | 0.9 |
| | 35–39 | 1.2 | 1.3 | 1.5 | 1.5 | 1.7 | 1.5 | 1.2 | 1.3 | 1.4 |
| | ≥ 40 | 1.8 | 1.9 | 2.4 | 2.3 | 2.4 | 2.0 | 1.6 | 1.7 | 1.8 |
| Antianemic preparations | **all** | **4.4** | **5.2** | **6.5** | **30.2** | **18.6** | **19.3** | **10.2** | **1.5** | **1.3** |
| | ≤ 34 | 3.6 | 4.4 | 5.7 | 31.0 | 18.3 | 19.5 | 9.8 | 1.4 | 1.3 |
| | 35–39 | 5.1 | 6.0 | 7.3 | 28.7 | 18.5 | 18.5 | 10.4 | 1.5 | 1.3 |
| | ≥ 40 | 6.8 | 7.5 | 9.0 | 29.2 | 20.7 | 19.6 | 12.2 | 2.0 | 1.5 |
| Drugs for acid related disorders | **all** | **3.2** | **3.2** | **3.1** | **2.7** | **2.3** | **2.5** | **1.5** | **1.6** | **1.8** |
| | ≤ 34 | 2.9 | 2.9 | 2.8 | 2.5 | 2.1 | 2.2 | 1.3 | 1.4 | 1.6 |
| | 35–39 | 3.7 | 3.6 | 3.4 | 3.0 | 2.6 | 2.7 | 1.7 | 1.7 | 2.0 |
| | ≥ 40 | 4.2 | 4.2 | 4.1 | 3.9 | 3.5 | 3.8 | 2.1 | 2.2 | 2.4 |
| Progestogens | **all** | **1.9** | **2.3** | **3.0** | **15.3** | **7.1** | **3.8** | **0.1** | **0.1** | **0.2** |
| | ≤ 34 | 1.1 | 1.4 | 1.7 | 12.4 | 5.7 | 3.6 | 0.1 | 0.1 | 0.2 |
| | 35–39 | 2.4 | 3.0 | 3.8 | 17.7 | 8.3 | 4.0 | 0.1 | 0.1 | 0.2 |
| | ≥ 40 | 5.3 | 6.1 | 8.8 | 27.9 | 12.0 | 4.9 | 0.1 | 0.1 | 0.3 |
| Gonadotropins | **all** | **1.0** | **1.3** | **2.4** | **1.2** | **0.0** | **0.0** | **0.0** | **0.0** | **0.0** |
| | ≤ 34 | 0.5 | 0.7 | 1.4 | 0.7 | 0.0 | 0.0 | 0.0 | 0.0 | 0.0 |
| | 35–39 | 1.5 | 1.9 | 3.7 | 1.8 | 0.0 | 0.0 | 0.0 | 0.0 | 0.0 |
| | ≥ 40 | 3.2 | 3.4 | 4.9 | 2.5 | 0.0 | 0.0 | 0.0 | 0.0 | 0.0 |
| Heparin preparations | **all** | **0.8** | **0.7** | **0.8** | **2.1** | **2.5** | **4.1** | **22.3** | **0.3** | **0.3** |
| | ≤ 34 | 0.6 | 0.5 | 0.5 | 1.2 | 1.6 | 3.0 | 17.8 | 0.2 | 0.3 |
| | 35–39 | 0.9 | 0.8 | 0.9 | 2.7 | 3.1 | 5.1 | 27.3 | 0.3 | 0.4 |
| | ≥ 40 | 1.7 | 1.7 | 2.4 | 6.4 | 6.0 | 8.6 | 37.3 | 0.4 | 0.4 |
| Antibiotics | **all** | **14.3** | **14.5** | **14.5** | **12.0** | **16.0** | **11.4** | **15.3** | **9.7** | **11.0** |
| | ≤ 34 | 13.9 | 14.1 | 14.2 | 11.0 | 14.0 | 11.4 | 15.2 | 9.6 | 10.9 |
| | 35–39 | 15.0 | 15.1 | 15.0 | 13.2 | 18.6 | 11.3 | 15.5 | 9.9 | 11.1 |
| | ≥ 40 | 15.3 | 15.4 | 15.2 | 15.0 | 21.6 | 11.6 | 15.9 | 9.6 | 10.8 |
| Anti-inflammatory drugs | **all** | **2.3** | **2.3** | **2.3** | **1.1** | **0.4** | **0.4** | **1.6** | **1.4** | **1.6** |
| | ≤ 34 | 2.1 | 2.1 | 2.0 | 1.0 | 0.4 | 0.3 | 1.4 | 1.3 | 1.4 |
| | 35–39 | 2.6 | 2.5 | 2.5 | 1.1 | 0.5 | 0.3 | 1.7 | 1.5 | 1.7 |
| | ≥ 40 | 3.2 | 3.2 | 3.0 | 1.5 | 0.6 | 0.5 | 2.3 | 2.0 | 2.1 |
| Corticosteroids | **all** | **2.5** | **2.4** | **2.6** | **2.1** | **1.2** | **1.7** | **1.3** | **1.5** | **1.8** |
| | ≤ 34 | 2,2 | 2.2 | 2.2 | 1.4 | 1.0 | 1.4 | 1.2 | 1.4 | 1.7 |
| | 35–39 | 2,7 | 2.7 | 2.9 | 2.4 | 1.4 | 1.9 | 1.4 | 1.6 | 2.0 |
| | ≥ 40 | 3,4 | 3.3 | 4.5 | 5.3 | 2.1 | 2.5 | 1.6 | 1.7 | 2.1 |
| Thyroid hormone | **all** | **2.7** | **2.8** | **2.9** | **4.4** | **5.8** | **5.1** | **3.2** | **3.0** | **3.1** |
| | ≤ 34 | 2.0 | 2.1 | 2.3 | 3.7 | 5.0 | 4.4 | 2.6 | 2.4 | 2.5 |
| | 35–39 | 3.4 | 3.5 | 3.7 | 5.3 | 6.6 | 5.8 | 3.8 | 3.6 | 3.7 |
| | ≥ 40 | 4.7 | 4.9 | 5.1 | 7.2 | 8.5 | 7.3 | 4.9 | 4.7 | 4.7 |
| Lipid modifying agents | **all** | **0.1** | **0.1** | **0.1** | **0.1** | **0.0** | **0.1** | **0.1** | **0.1** | **0.1** |
| | ≤ 34 | 0.1 | 0.1 | 0.1 | 0,1 | 0.0 | 0.1 | 0.1 | 0.1 | 0.1 |
| | 35–39 | 0.2 | 0.1 | 0.1 | 0,1 | 0.0 | 0.1 | 0.1 | 0.1 | 0.1 |
| | ≥ 40 | 0.2 | 0.2 | 0.3 | 0,2 | 0.1 | 0.1 | 0.1 | 0.2 | 0.2 |

(*Continued*)

**Table 2.** (Continued)

| Therapeutic groups | Age group | Pre-pregnancy | | | During pregnancy | | | Post-pregnancy | | |
|---|---|---|---|---|---|---|---|---|---|---|
| | | -III | -II | -I | I | II | III | +I | +II | +III |
| Antipertensives | **all** | **0.8** | **0.8** | **0.7** | **0.7** | **0.7** | **1.4** | **2.4** | **1.0** | **0.9** |
| | ≤ 34 | 0.5 | 0.5 | 0.4 | 0.4 | 0.5 | 1.1 | 1.7 | 0,6 | 0.6 |
| | 35–39 | 1.1 | 1.0 | 1.0 | 1.0 | 0.9 | 1.6 | 3.0 | 1,3 | 1.3 |
| | ≥ 40 | 1.9 | 1.9 | 1.8 | 1.8 | 1.7 | 2.8 | 5.3 | 2,6 | 2.4 |
| Antidiabetics | **all** | **0.5** | **0.5** | **0.5** | **0.6** | **1.0** | **2.1** | **0.3** | **0.3** | **0.4** |
| | ≤ 34 | 0.4 | 0.4 | 0.4 | 0.5 | 0.8 | 1.6 | 0.3 | 0.3 | 0.3 |
| | 35–39 | 0.5 | 0.5 | 0.5 | 0.6 | 1.3 | 2.7 | 0.3 | 0.3 | 0.4 |
| | ≥ 40 | 0.6 | 0.6 | 0.7 | 0.8 | 1.9 | 3.8 | 0.4 | 0.4 | 0.5 |
| Antiasthmatics | **all** | **3.4** | **3.4** | **3.3** | **2.6** | **3.1** | **2.5** | **2.0** | **2.4** | **2.7** |
| | ≤ 34 | 3.1 | 3.1 | 3.1 | 2.4 | 2.9 | 2.3 | 1.8 | 2.3 | 2.5 |
| | 35–39 | 3.9 | 3.9 | 3.7 | 2.9 | 3.4 | 2.9 | 2.2 | 2.6 | 3.0 |
| | ≥ 40 | 3.8 | 4.0 | 3.7 | 3.0 | 3.6 | 3.1 | 2.3 | 2.7 | 3.1 |
| Antiepileptics | **all** | **0.5** | **0.5** | **0.4** | **0.4** | **0.3** | **0.3** | **0.4** | **0.4** | **0.4** |
| | ≤ 34 | 0.5 | 0.4 | 0.4 | 0.3 | 0.3 | 0.3 | 0.3 | 0.4 | 0.4 |
| | 35–39 | 0.6 | 0.5 | 0.5 | 0.4 | 0.3 | 0.3 | 0.4 | 0.4 | 0.5 |
| | ≥ 40 | 0.6 | 0.6 | 0.5 | 0.4 | 0.3 | 0.3 | 0.4 | 0.5 | 0.5 |

The overall rate of medication prescriptions, as well as the prescription patterns observed in this large cohort of pregnant women are generally comparable with medication prescriptive trends observed in other European population-based studies; in particular, the reported rates ranged from 69.2 to 79.1% in the Netherlands [30], 81.8 to 89.3% in Belgium [31], 85.2 to 96% in Germany [32] and 89.9% in France [33], depending on whether the over-the-counter (OTC) medications are included or not in the analyses.

The peak of prevalence in the 1st TP was mainly due to a higher demand for some medications in early pregnancy, such as folic acid, iron preparations, vitamins and progestogens. In particular, the prevalence of folic acid seems to be very low during the preconception period and highest after pregnancy confirmation within the 1st TP, probably because women do not plan their pregnancy or do not request a preconception medical visit [34, 35]. The real consumption of folic acid in this study was likely underestimated due to the widespread use of antenatal non-reimbursed OTC medications and vitamin supplements in Italy [36]. The noteworthy low prescription rate in both pre-T and 1st TP periods showed a national or local clinical practice far from the recommendations of national and international clinical guidelines recommending a daily supplementation with 0.4 mg folic acid in women planning to become pregnant at least one month before the conception and until 12 weeks of gestation, in order to reduce the risk of neural tube defects and other congenital anomalies in their infants [37–41].

As regard to progestogens, a proportion of 19.0% of women was exposed to progesterone during pregnancy, mainly concentrated in the 1st TP (14.8%), probably in an attempt to prevent miscarriages. The use of progestogens, in particular in the prevention of non-recurrent miscarriage, is worthy of attention because its efficacy is still unsupported by conclusive evidence [42, 43]. In 2009 the World Health Organization (WHO) recommended not to prescribe progestogens for preventing miscarriages. In 2015 the American College of Obstetricians and Gynecologists (ACOG) Guideline stated that conclusive evidence supporting progestogens use to avoid early pregnancy loss is lacking and that women who have experienced at least three

**Table 3. The most prescribed 30 medications during pregnancy: Analysis by pregnancy trimester.**

| | ATC V level | Drug | Pregnancy | | I trimester | | II trimester | | III trimester | |
|---|---|---|---|---|---|---|---|---|---|---|
| | | | n | % | n | % | n | % | n | % |
| 1 | B03BB01 | folic acid | 155,233 | 34.6 | 131,102 | 29.2 | 56,490 | 12.6 | 31,220 | 7.0 |
| 2 | G03DA04 | progesteron | 85,224 | 19.0 | 66,601 | 14.8 | 26,905 | 6.0 | 13,446 | 3.0 |
| 3 | B03AA07 | ferrous sulfate | 84,206 | 18.8 | 9,340 | 2.1 | 35,331 | 7.9 | 60,326 | 13.5 |
| 4 | J01CR02 | amoxicillin/clavulanic acid | 51,495 | 11.5 | 17,252 | 3.8 | 21,282 | 4.7 | 18,459 | 4.1 |
| 5 | H03AA01 | levothyroxine | 34,399 | 7.7 | 19,931 | 4.4 | 25,822 | 5.8 | 22,641 | 5.1 |
| 6 | J01XX01 | fosfomycin | 32,301 | 7.2 | 9,771 | 2.2 | 14,865 | 3.3 | 11,007 | 2.5 |
| 7 | J01FA10 | azithromycin | 32,195 | 7.2 | 10,221 | 2.3 | 18,906 | 4.2 | 4,766 | 1.1 |
| 8 | J01CA04 | amoxicillin | 29,519 | 6.6 | 8,549 | 1.9 | 13,034 | 2.9 | 10,341 | 2.3 |
| 9 | R03BA01 | beclometasone | 20,760 | 4.6 | 6,391 | 1.4 | 8,906 | 2.0 | 7,094 | 1.6 |
| 10 | B01AB05 | enoxaparin | 18,131 | 4.0 | 7,264 | 1.6 | 8,623 | 1.9 | 14,402 | 3.2 |
| 11 | G03DA03 | hydroxyprogesterone | 12,923 | 2.9 | 5,783 | 1.3 | 7,014 | 1.6 | 4.524 | 1.0 |
| 12 | B01AC06 | low-dose aspirin | 11,782 | 2.6 | 7,870 | 1.8 | 7,842 | 1.7 | 3.449 | 0.8 |
| 13 | A11CC05 | colecalciferol | 11,151 | 2.5 | 4,904 | 1.1 | 5,548 | 1.2 | 4,648 | 1.0 |
| 14 | A02BX13 | alginate/bicarbonate | 10,016 | 2.2 | 3,805 | 0.8 | 4,425 | 1.0 | 4,948 | 1.1 |
| 15 | H02AB01 | betamethasone | 9,185 | 2.0 | 2,592 | 0.6 | 2,192 | 0.5 | 4,927 | 1.1 |
| 16 | B03AA01 | ferrous glycine sulfate | 8,477 | 1.9 | 1,008 | 0.2 | 3,360 | 0.7 | 5,922 | 1.3 |
| 17 | J01DD08 | cefixime | 8,237 | 1.8 | 2,846 | 0.6 | 3,060 | 0.7 | 2.784 | 0.6 |
| 18 | G03CA03 | estradiol | 7,950 | 1.8 | 7,934 | 1.8 | 413 | 0.1 | 25 | 0.0 |
| 19 | H02AB07 | prednisone | 7,801 | 1.7 | 5,685 | 1.3 | 2,632 | 0.6 | 1,997 | 0.4 |
| 20 | A02AD02 | magaldrate | 5,891 | 1.3 | 2,563 | 0.6 | 2,066 | 0.5 | 2,065 | 0.5 |
| 21 | J06BB01 | human immunoglobulin anti-D (Rh0) | 5,812 | 1.3 | 349 | 0.1 | 1,401 | 0.3 | 4,238 | 0.9 |
| 22 | J01FA09 | clarithromycin | 5,636 | 1.3 | 2,342 | 0.5 | 1,799 | 0.4 | 1,715 | 0.4 |
| 23 | R03AC02 | salbutamol | 5,255 | 1.2 | 2,153 | 0.5 | 2,331 | 0.5 | 1,791 | 0.4 |
| 24 | A10AE05 | insulin detemir | 4,999 | 1.1 | 409 | 0.1 | 1,775 | 0.4 | 3,899 | 0.9 |
| 25 | J01CA01 | ampicillin | 4,778 | 1.1 | 988 | 0.2 | 2,163 | 0.5 | 1,841 | 0.4 |
| 26 | B01AB06 | nadroparin | 4,654 | 1.0 | 1,980 | 0.4 | 2,131 | 0.5 | 3,479 | 0.8 |
| 27 | C08CA05 | nifedipine | 3,801 | 0.8 | 705 | 0.2 | 1,308 | 0.3 | 3,209 | 0.7 |
| 28 | A10AB04 | insulin lispro | 3,405 | 0.8 | 654 | 0.1 | 1,414 | 0.3 | 2,806 | 0.6 |
| 29 | J01MA02 | ciprofloxacin | 3,316 | 0.7 | 2,081 | 0.5 | 671 | 0.1 | 633 | 0.1 |
| 30 | A05AA02 | ursodeoxycholic acid | 3,079 | 0.7 | 193 | <0.05 | 619 | 0.1 | 2,757 | 0.6 |

* denominator: pregnancies reaching the 3rd TP (exclusion of deliveries occurred between 20–27 weeks of gestation)

prior miscarriages may benefit from progesterone therapy in the first trimester [44]. https://www.nejm.org/doi/full/10.1056/NEJMoa1813730?url_ver=Z39.88-2003&rfr_id=ori:rid:crossref.org&rfr_dat=cr_pub%20%200pubmedThis recommendation is supported by the results of a large multicentre randomized clinical trial conducted on use of progesterone to prevent miscarriage in women with vaginal bleeding in early pregnancy [45]. The persistence of this inappropriate prescriptive habit in Italy makes the monitoring of medication prescriptions in this therapeutic area helpful to promote appropriate use.

In our study antibiotics for systemic use were the second most prescribed medication during pregnancy (peak of 16% in 2nd TP), even if the observed rate is lower than those found in other European countries (27–40%) [31, 33, 46–48]. Moreover, eight different antibiotics were among the 30 most prescribed drugs. Given the growing challenge of bacterial resistance, and the increased risk of some adverse neonatal outcomes in women exposed during pregnancy, albeit possibly due to the infections itself, the use of antibiotics during pregnancy requires

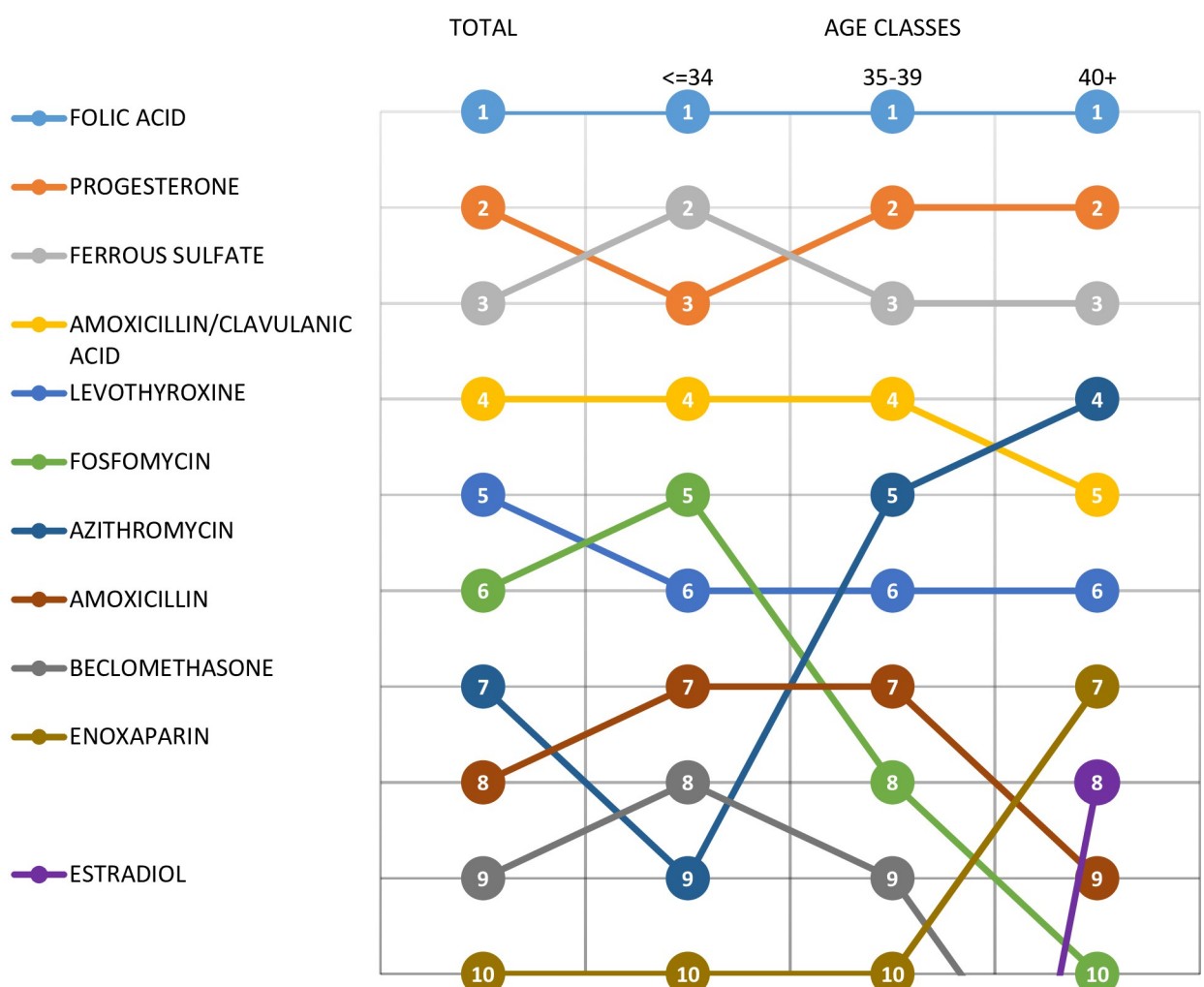

**Fig 3. Ranking of the most 10 prescribed drugs during pregnancy overall and by maternal age group.**

attention [49] because sepsis is one of the leading global causes of maternal and neonatal death [50]. The peak of antibiotic prescription rate detected in the 2nd TP is probably related to amniocentesis although recommendations in favor of antibiotic prophylaxis in women undergoing invasive prenatal diagnosis are lacking [51]. The routine screening for asymptomatic bacteriuria during the first trimester, and for Group B Beta-Haemolytic Streptococcus together with antibiotic prophylaxis in case of caesarean section reasonably contribute to the increase in antibiotic prescriptions during early and late pregnancy.

The increase in heparin preparation prevalence detected among all age groups reaching a peak of 37.3% in women aged ≥40 years in the first post-partum trimester could be explained by the clinical characteristics of the older women (e.g. risk factors for venous thromboembolism, such as hypertension, cardiovascular diseases and obesity) or by thromboprophylaxis for women undergoing caesarean section, as recommended by the international guidelines [52–56]. This trend, especially the large use during pregnancy in women>40 years (8.6%) need to be investigated thoroughly by maternal clinical characteristics in order to evaluate the appropriate use of these medications.

The increase of antihypertensive, antidiabetics and thyroid hormone prescriptions observed during pregnancy is probably associated to gestational hypertension, diabetes and hypothyroidism. On the contrary, the decrease in prescriptions observed for chronic therapies such as antiepileptics or lipid-modifying agents during pregnancy, is likely related to the possible negative effects of these medications on the fetus. More interventions to inform women and clinicians on the related risk-benefits profile of these medications are needed to protect the health of both mother and fetus.

The strength of our study is the availability of the medication prescription data in pregnancy from eight different regions, which are representative of all Italian geographical areas. To our knowledge, this is the largest and most representative population-based study illustrating the medication prescription during pregnancy in Italy. Previous population-based studies conducted in Italy were limited to single Italian regions [22–25] or smaller samples [10].

A limitation of the study is that our data were referred only to prescription of medicines reimbursed by the Italian National Health Service, excluding OTC and non-reimbursed medications (i.e. vitamin supplements), that may lead to an underestimation of medication use among the target population. Finally, our administrative databases do not provide information on drug use in pregnancies ended in miscarriage or induced abortion, as well as no information on therapeutic indications for drug prescribing were available, consequently we were not able to fully investigate the medication use patterns.

## Conclusions

The evaluation and monitoring of medicine prescriptions in the preconception period, during pregnancy and after childbirth are aspects of extreme interest in term of public health [57]. It is therefore important to have a national observatory monitoring the prescribing trends of drugs for women of childbearing age, during pregnancy and the post-partum period. The first study conducted within MoM-Net has made available an exhaustive overview of the medication use during pregnancy in Italy through the integration of multiple regional health databases. The periodic replication of this study involving more regions up to national coverage, will achieve the objectives of monitoring the critical aspects of drug prescribing among pregnant women, and to detect medicine prescriptions at risk of inappropriate use.

## Acknowledgments

**MoM-Net group**: Filomena Fortinguerra, Serena Perna, Francesco Trotta, (Italian Medicines Agency, Rome); Renata Bortolus, Giovanni Rezza (Directorate General for Preventive Health—Office 9, Ministry of Health, Rome); Paola D'Aloja, Serena Donati (National Centre for Disease Prevention and Health Promotion, Istituto Superiore di Sanità, Rome); Roberto Da Cas (Pharmacoepidemiology Unit, National Centre for Drug Research and Evaluation, Istituto Superiore di Sanità, Rome); Antonio Addis, Valeria Belleudi, Marina Davoli, Lorella Lombardozzi, Francesca Romana Poggi (Department of Epidemiology, Lazio Regional Health Service, ASL Roma 1, Rome); Antonio Clavenna (Laboratory for Pharmacoepidemiology, Department of Public Health, IRCCS–Istituto di Ricerche Farmacologiche Mario Negri, Milan, Italy); Anna Locatelli (Obstetrics and Gynecology Unit, School of Medicine and Surgery, University of Milano Bicocca, Milan); Ida Fortino, Arianna Mazzone, Simone Schiatti, Martina Zanforlini, (Lombardy Region); Paola Deambrosis, Silvia Manea, Laura Salmaso, Giovanna Scroccaro (Veneto Region), Anna Maria Marata, Aurora Puccini, Valentina Solfrini, (Emilia-Romagna Region); Francesco Attanasio, Rosa Gini (Tuscany Region); Marcello De Giorgi, David Franchini, Mariangela Rossi (Umbria Region); Vito Montanaro Paolo Stella (Puglia Region); Paolo Carta, Donatella Garau, Stefano Ledda, Enrico Serra (Sardinia Region).

**Disclaimer:** The views expressed in this article are the personal views of the authors and may not be understood or quoted as being made on behalf of or reflecting the position of the respective authors' organization.

## Author Contributions

**Conceptualization:** Filomena Fortinguerra, Valeria Belleudi, Francesco Trotta.

**Data curation:** Valeria Belleudi, Francesca Romana Poggi.

**Formal analysis:** Valeria Belleudi, Francesca Romana Poggi, Serena Perna.

**Investigation:** Filomena Fortinguerra, Renata Bortolus, Serena Donati, Paola D'Aloja, Antonio Clavenna, Anna Locatelli.

**Methodology:** Valeria Belleudi.

**Project administration:** Filomena Fortinguerra.

**Supervision:** Filomena Fortinguerra, Valeria Belleudi, Francesco Trotta.

**Validation:** Valeria Belleudi, Renata Bortolus, Serena Donati, Paola D'Aloja, Antonio Clavenna, Anna Locatelli.

**Visualization:** Valeria Belleudi, Francesca Romana Poggi, Serena Perna.

**Writing – original draft:** Filomena Fortinguerra, Antonio Clavenna.

**Writing – review & editing:** Filomena Fortinguerra, Valeria Belleudi, Francesca Romana Poggi, Serena Perna, Renata Bortolus, Serena Donati, Paola D'Aloja, Roberto Da Cas, Anna Locatelli, Antonio Addis, Marina Davoli, Francesco Trotta.

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
