## [Decision Letter · Decision Letter 0]

9 Jan 2023

PONE-D-22-25133Monitoring medicine prescriptions before, during and after pregnancy in ItalyPLOS ONE

Dear Dr. Fortinguerra,

Thank you for submitting your manuscript to PLOS ONE. After careful consideration, we feel that it has merit but does not fully meet PLOS ONE’s publication criteria as it currently stands. Therefore, we invite you to submit a revised version of the manuscript that addresses the points raised during the review process, especially the comments about the confounding variance. 

We look forward to receiving your revised manuscript.

Kind regards,

Linglin Xie

Academic Editor

PLOS ONE

Journal Requirements:

Reviewers' comments:

Reviewer's Responses to Questions

**Comments to the Author**

1. Is the manuscript technically sound, and do the data support the conclusions?

Reviewer #1: Partly

2. Has the statistical analysis been performed appropriately and rigorously? 

Reviewer #1: N/A

3. Have the authors made all data underlying the findings in their manuscript fully available?

Reviewer #1: Yes

4. Is the manuscript presented in an intelligible fashion and written in standard English?

Reviewer #1: Yes

5. Review Comments to the Author

Reviewer #1: I was kindly asked to review the manuscript entitled “Monitoring medicine prescriptions before, during and after pregnancy in Italy” by Fortinguerra F et al. The aim of the study was to describe the clinical attitude about drug prescription in Italy, with regard to one of the most vulnerable population, id est pregnant women, and in the period before and after pregnancy.

Results are interesting and well mirror clinical practice, but, given that it should be conducted by an evidence-based approach and in light of clinical practice guidelines, it makes results not so novel and surprising as well.

Surely, authors put many efforts in this study and the reason behind is of relevance, but, as ascertained, its only strength regards the large sample size. In my personal opinion, this study presents the following main limitation. The analysis is not adjusted by confounding variables such as race, BMI and type of pregnancy (single, twin, natural conception or IVF, etc…). Furthermore, It should be also valuable to investigate the relationship, if any, between the incidence of the most meaningful drugs and pregnancy outcomes in mothers and fetuses.

Before publication, I suggest authors to consider my comment if suitable to improve their study, if suitable to improve their study, and the following minor points:

- line 55: TP abbreviation needs to be defined;

- line 58: “…≥40” should be followed by “years”;

- line 127: This sentence does not make sense, rephrase or delete.

6. PLOS authors have the option to publish the peer review history of their article (what does this mean?). If published, this will include your full peer review and any attached files.

Reviewer #1: No

---

## [Author Response · Author response to Decision Letter 0]

14 Mar 2023

Journal Requirements:

Authors’ answer: we checked the manuscript and confirm that our manuscript meets all PLOS ONE's style requirements 

Authors’ answer: we added the details on participant consent reported in Ethics statement also in the Methods section of the manuscript. 

Authors’ answers: the corresponding author have a validated ORCID ID in Editorial Manager. 

Authors’ answer: we updated the Data Availability statement in the manuscript. 

 

5. Review Comments to the Author

Reviewer #1: I was kindly asked to review the manuscript entitled “Monitoring medicine prescriptions before, during and after pregnancy in Italy” by Fortinguerra F et al. The aim of the study was to describe the clinical attitude about drug prescription in Italy, with regard to one of the most vulnerable population, id est pregnant women, and in the period before and after pregnancy.

Results are interesting and well mirror clinical practice, but, given that it should be conducted by an evidence-based approach and in light of clinical practice guidelines, it makes results not so novel and surprising as well.

Surely, authors put many efforts in this study and the reason behind is of relevance, but, as ascertained, its only strength regards the large sample size. In my personal opinion, this study presents the following main limitation. The analysis is not adjusted by confounding variables such as race, BMI and type of pregnancy (single, twin, natural conception or IVF, etc…). Furthermore, It should be also valuable to investigate the relationship, if any, between the incidence of the most meaningful drugs and pregnancy outcomes in mothers and fetuses.

Before publication, I suggest authors to consider my comment if suitable to improve their study, if suitable to improve their study, and the following minor points:

- line 55: TP abbreviation needs to be defined;

- line 58: “…≥40” should be followed by “years”;

- line 127: This sentence does not make sense, rephrase or delete.

 

Authors’ response to Reviewer #1

Dear reviewer, thank you for your comments, which give us the opportunity to better clarify the objectives and implications of our study. 

We well know that the outcomes on health status of both women and newborns of medication use during pregnancy is a topic of great interest in the field of public health. However, we believe that monitoring medicine prescriptions in clinical practice is a necessary step in assessing the impact of therapeutic choices in pregnant women as well as the adherence to the available clinical guidelines. Despite some limitations of our study (like not collecting information on outcomes of drug use in pregnancy or not adjusting by confounding variables such as race, BMI and type of pregnancy), its added value is to compare the prescriptive trends and patterns in Italy to those reported in other European countries and to identify any critical aspects in the current clinical practice first; in our opinion these elements are essential for the medical community in order to improve the current medical care for pregnant and childbearing woman on the basis of the already available evidence and clinical guidelines. 

Since we did not investigate the correlation between the incidence of the most meaningful drugs and pregnancy outcomes in mothers and fetuses, we did not adjust our analysis by confounding variables such as race, BMI and type of pregnancy (single, twin, natural conception or IVF, etc…). If we only evaluate the prescriptions and adherence to evidence-based clinical guideline why do we think that an obese woman should receive a different prescription? However, we have investigated the effect of the maternal age on drug prescribing profile. 

Based on these assumptions and given the limited information on medications currently used in Italian pregnant women, we performed a large population-based study, accounting about 59% of total births occurred in Italy during the study period, to provide a representative and updated overview of drug prescribing before, during and after pregnancy in Italy. 

The novelty of this study is that it is promoted by a regulatory agency, which set up a Working Group called MoM-Net (Monitoring Medication Use During Pregnancy Network) involving a network of eight Italian regions and a number of experts from Italian public and academic institutions, focused on a very delicate issue, such as the monitoring drug use during pregnancy by integrating various regional health databases; the objective is the periodically monitoring of the prescription patterns of drugs supplied by the Italian National Health Service to pregnant women, not only during the gestational period, but also in pre-conception and post-pregnancy. Also to estimate the treatment dropout rate is important. With this study we want to underline the importance of the network to investigate drug use in pregnancy, which can be enlarged. The data produced were robust and consistent on some information, testing the importance of the use of the information flows within a wide network to answer to emerging health issue in the field. Any issues related to clinically inappropriate prescribing, non-adherence to treatment or drug discontinuation due to pregnancy, switches analysis (replacement of a non-recommended drug in pregnancy with another considered safer or transition from poly-therapy to monotherapy), heterogeneity in prescriptive medical habits between Italian Regions and subgroups of populations (foreign women and women with births multiple) could be also investigated. 

The added value of these study results is also to hypothesize and propose lines for future research in the field. A more in-depth studies could be conducted for some categories of drugs for which no safety information is available on pregnancy and for which investigating the health outcomes of the pregnancy use or non-use of medications is essential in order to guarantee an optimal medical assistance. 

The integration of different regional databases, together with the sharing of data, has made it possible to produce sufficiently representative results and to obtain an exhaustive overview of the use of drugs in pregnancy in Italy, while emphasizing the importance of periodic monitoring of the information flows produced, training and information aimed at both prescribers (and pharmacists) and women on the appropriate use of drugs in the context of assistance to women of childbearing age, pregnancy and post-partum.

In addition, these results can represent a valid tool for the prescribers who wants to become aware of the real use of the drug in pregnancy. This approach, although tiring, represents a moment of professional growth for the prescribers who, in addition to measuring his own activity, produces robust data useful for the scientific community, strategic directions and colleagues. In addition, we think that our results provide useful information to identify evidence and research gaps or inappropriate clinical practice, on the basis of which to plan training activities and/or information interventions on the appropriate use of medicines for both prescribers (and pharmacists) and childbearing women. 

Finally, we found that other authors have already published studies like this on PLOS One, therefore we hope that our study will also be accepted for publication by the journal. Thank you. 

As suggested by the reviewer, we checked the following lines in the manuscript: 

- line 55: TP abbreviation needs to be defined;

We have replaced the acronym TP with the words “trimester of pregnancy”. Thank you

- line 58: “…≥40” should be followed by “years”;

We added the word. Thank you

- line 127: This sentence does not make sense, rephrase or delete.

We deleted the sentence. Thank you

---

## [Decision Letter · Decision Letter 1]

5 May 2023

PONE-D-22-25133R1Monitoring medicine prescriptions before, during and after pregnancy in ItalyPLOS ONE

Dear Dr. Fortinguerra,

Thank you for submitting your manuscript to PLOS ONE. After careful consideration, we feel that it has merit but does not fully meet PLOS ONE’s publication criteria as it currently stands. Therefore, we invite you to submit a revised version of the manuscript that addresses the points raised during the review process.

We look forward to receiving your revised manuscript.

Kind regards,

Linglin Xie

Academic Editor

PLOS ONE

Journal Requirements:

Reviewers' comments:

Reviewer's Responses to Questions

**Comments to the Author**

1. If the authors have adequately addressed your comments raised in a previous round of review and you feel that this manuscript is now acceptable for publication, you may indicate that here to bypass the “Comments to the Author” section, enter your conflict of interest statement in the “Confidential to Editor” section, and submit your "Accept" recommendation.

Reviewer #2: (No Response)

2. Is the manuscript technically sound, and do the data support the conclusions?

Reviewer #2: Yes

3. Has the statistical analysis been performed appropriately and rigorously? 

Reviewer #2: Yes

4. Have the authors made all data underlying the findings in their manuscript fully available?

Reviewer #2: Yes

5. Is the manuscript presented in an intelligible fashion and written in standard English?

Reviewer #2: Yes

6. Review Comments to the Author

Reviewer #2: Table 1: The sums of percentages provided in table for demographic characteristics are more than 100% in some cases e.g., age distribution.

7. PLOS authors have the option to publish the peer review history of their article (what does this mean?). If published, this will include your full peer review and any attached files.

Reviewer #2: No

---

## [Author Response · Author response to Decision Letter 1]

10 May 2023

Journal Requirements:

Authors’ answer: We checked our reference list to ensure that it is complete and correct. No retracted article were found. 

6. Review Comments to the Author

Reviewer #2: Table 1: The sums of percentages provided in table for demographic characteristics are more than 100% in some cases e.g., age distribution.

Authors’ answer: We thank the reviewer for the comment. We checked and revised the numbers and the percentages reported in Table 1, as suggested. We have uploaded the revised version of the manuscript, accordingly.

---

## [Editor Report · Decision Letter 2]

30 May 2023

Monitoring medicine prescriptions before, during and after pregnancy in Italy

PONE-D-22-25133R2

Dear Dr. Fortinguerra,

We’re pleased to inform you that your manuscript has been judged scientifically suitable for publication and will be formally accepted for publication once it meets all outstanding technical requirements.

Kind regards,

Linglin Xie

Academic Editor

PLOS ONE
---

## [Editor Report · Acceptance letter]

5 Jun 2023

PONE-D-22-25133R2 

Monitoring medicine prescriptions before, during and after pregnancy in Italy 

Dear Dr. Fortinguerra:

I'm pleased to inform you that your manuscript has been deemed suitable for publication in PLOS ONE. Congratulations! Your manuscript is now with our production department. 

Kind regards, 

on behalf of

Dr. Linglin Xie 

Academic Editor

PLOS ONE